# Diagnostic accuracy of contrast-enhanced CT for neck abscesses: A systematic review and meta-analysis of positive predictive value

Jon Hagelberg[1], Bernd Pape[2,3], Jaakko Heikkinen[4], Janne Nurminen[4], Kimmo Mattila[4], Jussi Hirvonen[1,4]*

1 Department of Radiology, Faculty of Medicine and Health Technology and Tampere University Hospital, Tampere University, Tampere, Finland, 2 Turku Clinical Research Center, Turku University Hospital, Turku, Finland, 3 School of Technology and Innovations, University of Vaasa, Vaasa, Finland, 4 Department of Radiology, University of Turku and Turku University Hospital, Turku, Finland

* jussi.hirvonen@utu.fi

**Data Availability Statement:** All relevant data are within the paper and its Supporting Information files.

## Abstract

### Objectives

To review the diagnostic accuracy of contrast-enhanced computed tomography (CT) in differentiating abscesses from cellulitis in patients with neck infections, using surgical findings as the reference standard.

### Materials and methods

Previous studies in the last 32 years were searched from PubMed and Embase. Because of partial verification bias (only positive abscess findings are usually verified surgically), sensitivity and specificity estimates are unreliable, and we focused on positive predictive value (PPV). For all studies, PPV was calculated as the proportion of true positives out of all positives on imaging. To estimate pooled PPV, we used both the median with an interquartile range and a model-based estimate. For narrative purposes, we reviewed the utility of common morphological CT criteria for abscesses, such as central hypodensity, the size of the collection, bulging, rim enhancement, and presence of air, as well as sensitivity and specificity values reported by the original reports.

### Results

23 studies were found reporting 1453 patients, 14 studies in children (771 patients), two in adults (137 patients), and seven including all ages (545 patients). PPV ranged from 0.67 to 0.97 in individual studies, had a median of 0.84 (0.79–0.87), and a model-based pooled estimate of 0.83 (95% confidence interval 0.80–0.85). Most morphological CT criteria had considerable overlap between abscesses and cellulitis.

**Funding:** Sigrid Juséliuksen Säätiö http://dx.doi.org/10.13039/501100006306 The funders had no role in study design, data collection and analysis, decision to publish, or preparation of the manuscript.

**Competing interests:** The authors have declared that no competing interests exist.

## Conclusions

The pooled estimate of PPV is 0.83 for diagnosing neck abscesses with CT. False positives may be due to limited soft tissue contrast resolution. Overall, none of the morphological criteria seem to be highly accurate for differentiation between abscess and cellulitis.

## Introduction

The purposes of imaging in deep neck infections are to diagnose and localize drainable abscesses, provide differential diagnosis, and assess potential complications, such as venous thrombosis, mediastinal involvement, and airway compromise. Contrast-enhanced computed tomography (CT) is considered the standard imaging modality because of its good availability and low cost [1]. However, the diagnostic accuracy of CT varies substantially between studies, and several diagnostic criteria exist for the diagnosis of an abscess, including low-density core, rim enhancement, bulging, or scalloping [2]. Especially in early abscess formation, these diagnostic signs may be subtle. The purpose of this study is to review previous CT studies on the diagnostic accuracy for abscesses in patients with neck infections and critically review several diagnostic criteria.

Studies into diagnostic accuracy in detecting abscesses using imaging are hampered by biased estimates of sensitivity and specificity. This is for the most part due to inherent difficulties in ascertaining the reference standard. The reference standard for an abscess is most often surgical proof of purulence, but typically only positive cases (those with imaging evidence of abscess) undergo surgery. This problem is referred to as partial verification bias, in which only one result of an index test is verified using the reference standard [3]. Because the proportions of true and false negatives are unknown, partial verification bias will tend to overestimate the sensitivity and underestimate specificity. In the case of neck abscesses, true negatives are often inferred as patients who recover uneventfully after conservative treatment. While this reference standard is reasonable in clinical practice, small abscesses may be included as false negatives. This approach leads to differential verification bias [3] and may not effectively mitigate biased estimation of sensitivity and specificity.

Here, we review evidence for the diagnostic accuracy of CT in diagnosing neck abscesses. Due to partial and/or differential verification bias and ensuing problems of correctly estimating sensitivity and specificity, this systematic review and meta-analysis focuses on positive predictive value (PPV). PPV is the proportion of patients with a positive reference test (abscess in surgery) among patients with a positive index test (abscess on CT), in other words, the proportion of true positives among all positives. PPV is not affected by partial verification bias, because only surgically treated patients are evaluated in terms of whether purulence was encountered or not. In the setting of neck infections, PPV is clinically significant, because false positives may lead to unnecessary surgery. For narrative purposes, we also evaluate sensitivity and specificity estimates reported by original publications.

## Methods

In this systematic review and meta-analysis, we followed the Preferred Reporting Items for Systematic Reviews and Meta-Analyses (PRISMA) guideline [4]. We searched PubMed and Embase for the terms "neck infection" OR "neck infections". The search protocol was not registered beforehand. In many clinical case series and retrospective cohort studies, imaging is

not reported in the abstract or the keywords. Therefore, we did not include "computed tomography", "CT", or "imaging" in the search terms. We used the following inclusion criteria: 1) study published in the English language between 1/1/1990 and 8/31/2022; 2) conventional single-energy, monophasic, single-scan contrast-enhanced CT diagnosis of presence or absence of a neck abscess; 3) surgical confirmation of presence or absence of purulence; 4) numbers needed to estimate PPV are unequivocally reported: total number of patients with a positive CT finding undergoing operative treatment (needle aspiration, incision and drainage, or open surgery), stratified into those who had purulence and those who did not. We used the following exclusion criteria: 1) 20 or fewer CT-positive patients with surgery; non-conventional CT methodology (such as biphasic contrast administration, or dual-energy acquisition; and 3) history of radiation therapy of the neck. To be included, studies had to specifically report having included "abscesses" and not simply "infections", because the latter might include non-purulent conditions such as phlegmon/cellulitis. In addition, preoperative diagnoses had to be unequivocal. Additional entries were extracted from reference lists of included articles. Duplicates were manually excluded. Applicability and risk of bias were assessed using QUADAS-2 [5] with the following criteria: whether patients were randomly chosen (risk of bias, patient selection), whether CT criteria for abscess were defined (risk of bias, index test), whether surgery was done within 48 h of imaging or whether all patients with abscesses on CT had surgery (risk of bias, flow, and timing), whether both children and adults were included (applicability concerns, patient selection). The risk was considered low for all studies related to the reliability of surgical reports (risk of bias, reference standard), the similarity of CT devices (applicability concerns, index test), and interpretation of surgical reports (applicability concerns, reference standard). Data review was carried out by the first author (J.Ha.). In addition, three board-certified radiologists (J.He., J.N., J.Hi.) independently confirmed study selection, data extraction, and study quality.

For a primary outcome measure, we extracted and tabulated the number of patients with a positive finding for an abscess on both CT finding and surgical exploration, and further divided that number into true and false positives to calculate PPV. According to a previously published procedure [6], we calculated the pooled estimate for PPV using two methods: 1) simple descriptive statistics including median and interquartile range and 2) random effects modeling of PPV using Proc Mixed on SAS System, version 9.4 for Windows (SAS Institute Inc., Cary, NC, USA) as previously described [7]. Meta-regression analyses were carried out with the following moderators: whether CT criteria of abscess were defined (yes/no), whether the size of the collection was considered (yes/no), whether rim enhancement was being analyzed (yes/no), whether the study was done only in children (yes/no), and age of publication (years, 2022 minus year of publication). Heterogeneity was assessed with the Higgins inconsistency test ($I^2$). For a secondary outcome measure, and only for narrative purposes, we extracted and tabulated sensitivity and specificity estimates for overall detection of an abscess, when and as explicitly reported in primary articles. A statistician (B.P.) performed or oversaw all statistical analyses.

## Results

We found 23 studies reporting 1453 patients (Fig 1, Table 1) [8–30]. Two of the studies (9%) were prospective, whereas 20 studies (87%) were retrospective, and one study did not report the study design. Of included studies, 14 (61%) were done in children (reporting 771 patients, 53% of total), two (9%) in adults (reporting 137 patients, 9% of total), and 7 (30%) including all ages (reporting 545 patients, 38% of total). Five studies (22%) focused exclusively on retropharyngeal abscesses [12, 18, 19, 27, 28], one study on lateral neck abscesses [15], and others

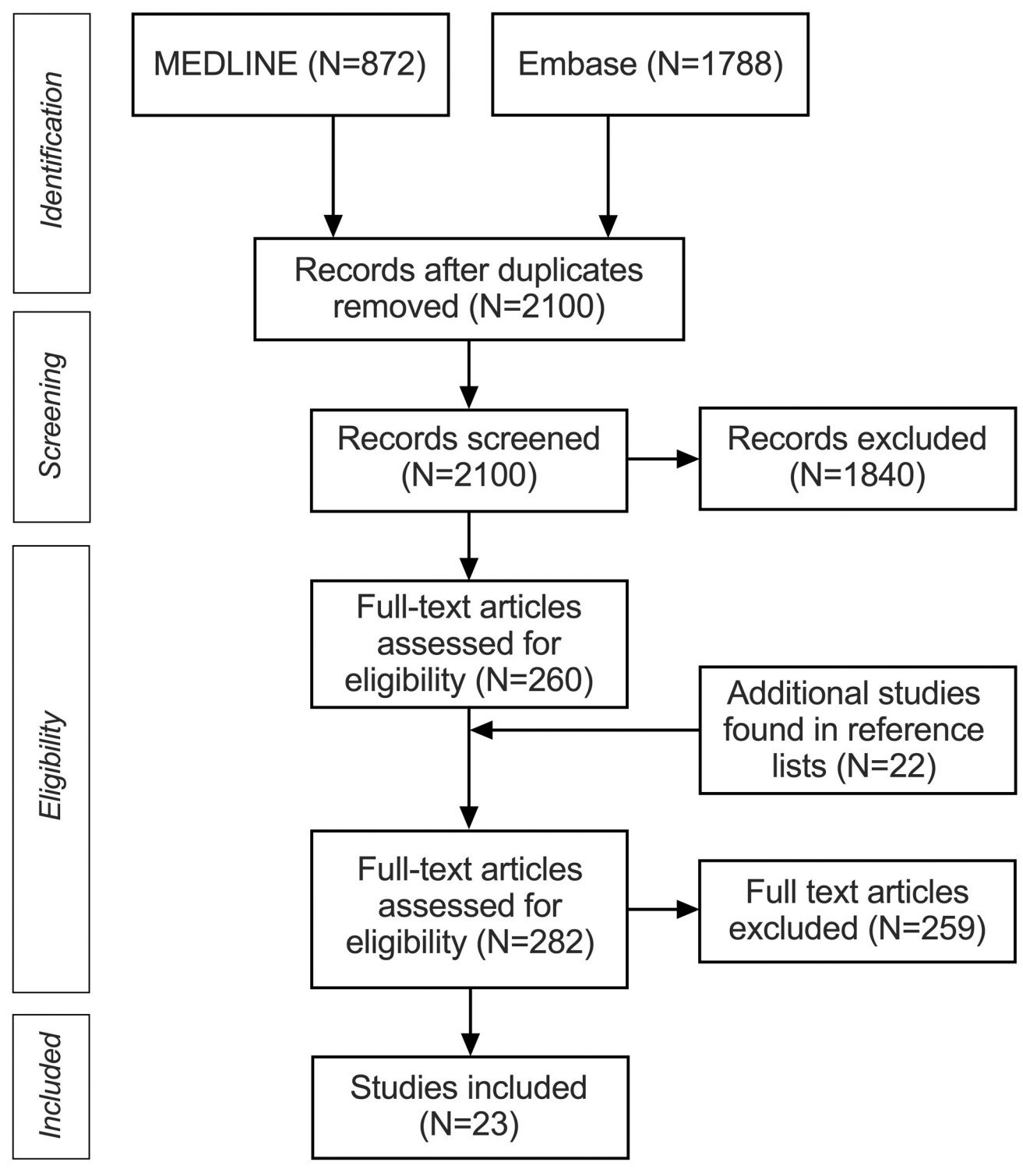

**Fig 1. PRISMA flowchart of systematic review and meta-analysis.**

included abscesses in various locations. A total of 19 studies were excluded due to the small sample size, reporting 206 patients. The quality assessment indicated risks of bias related to index test and flow of timing and applicability concerns related to patient selection (S1 Table).

**Table 1. Studies included in the meta-analysis.**

| First author | Year | N | Prim | Design | Patients | Age | Location | TP | FP | PPV | Se | Sp |
|---|---|---|---|---|---|---|---|---|---|---|---|---|
| Chuang | 2013 | 162 | yes | retrospective | all ages | 49.7 (1–93) | multiple | 129 | 33 | 0.796 | - | - |
| Wang | 2003 | 134 | no | retrospective | all ages | 41.8 (1–86) | multiple | 114 | 20 | 0.851 | - | - |
| Côrte | 2017 | 111 | no | retrospective | children | 7.73 | multiple | 87 | 24 | 0.784 | - | - |
| Boscolo-Rizzo | 2012 | 111 | no | retrospective | adults | 52[a] (18–96) | multiple | 97 | 14 | 0.874 | - | - |
| Page | 2008 | 97 | no | retrospective | children | 4.9 (0–17) | RPA | 79 | 18 | 0.814 | 0.72 | 0.59 |
| Elden | 2001 | 90 | yes | retrospective | children | N.R. (0.1–17) | multiple | 72 | 18 | 0.800 | - | - |
| Seer Yee | 2014 | 75 | yes | N.R. | all ages | 28.3 (0–98) | multiple | 71 | 4 | 0.947 | 0.99 | 0.67 |
| Collins | 2014 | 70 | yes | retrospective | children | 3.2 (0.1–18) | lateral | 61 | 9 | 0.871 | 0.68 | 0.18 |
| Freling | 2009 | 65 | yes | retrospective | all ages | 33 (1.5–83) | multiple | 53 | 12 | 0.815 | - | - |
| Meyer | 2009 | 64 | no | retrospective | children | 1 (0.5–17) | multiple | 59 | 5 | 0.922 | - | - |
| Kirse | 2001 | 62 | yes | retrospective | children | N.R. (0.2–15) | RPA | 50 | 12 | 0.806 | - | - |
| Hoffman | 2011 | 57 | no | retrospective | children | 1 (0.5–13.5) | RPA | 38 | 19 | 0.667 | - | - |
| Choi | 1997 | 45 | no | retrospective | children | 3.5 (0.1–17) | multiple | 34 | 11 | 0.756 | 0.75 | - |
| Ban | 2018 | 43 | no | retrospective | all ages | 41.5 | multiple | 39 | 4 | 0.907 | - | - |
| Wong | 2012 | 37 | no | retrospective | children | 4.8 (0.3–14) | multiple | 36 | 1 | 0.973 | - | - |
| Lazor | 1994 | 34 | yes | retrospective | all ages | 32.9 (1–75) | multiple | 29 | 5 | 0.853 | 0.88 | - |
| Malloy | 2008 | 32 | yes | retrospective | children | 3[a] | multiple | 28 | 4 | 0.875 | - | - |
| Smith | 2006 | 32 | yes | retrospective | all ages | N.R. (1–89) | multiple | 24 | 8 | 0.750 | - | - |
| Miller | 1999 | 26 | yes | prospective | adults | 37.8 (18–80) | multiple | 19 | 7 | 0.731 | 0.95 | 0.53 |
| Saluja | 2013 | 34 | no | prospective | children | 5.2 | RPA | 29 | 5 | 0.853 | - | - |
| Stone | 1999 | 25 | yes | retrospective | children | N.R. | RPA | 21 | 4 | 0.840 | 0.81 | 0.63 |
| Vural | 2003 | 24 | yes | retrospective | children | 4.9 (0.3–14) | multiple | 17 | 7 | 0.708 | 0.68 | 0.56 |
| Kurzyna | 2015 | 23 | no | retrospective | children | 4.9 (0–17) | multiple | 22 | 1 | 0.957 | - | - |

Age given as mean or median[a] (range, if available); N, number of patients who had abscess on CT and surgery; Prim, study designed to primarily investigate diagnostic accuracy; RPA, retropharyngeal abscess; TP, true positive; FP, false positive; PPV, positive predictive value; Se, sensitivity; Sp, specificity.

PPV ranged from 0.67 to 0.97 (Fig 2). Visual inspection of the funnel plot, examining PPV as a function of sample size, did not suggest any significant publication bias (Fig 3). Median PPV was 0.84 (IQR 0.79–0.87). Model-based pooled estimate of PPV was 0.83 (95% confidence interval 0.80–0.85). We found statistically significant evidence for heterogeneity among studies ($I^2$ = 43%, p = 0.007). Although PPV did not significantly correlate with sample size, studies with smaller sample sizes tended to show larger variation in PPV. For studies including more than 90 patients, PPV ranged from 0.78–0.85 and had a median of 0.81 (IQR 0.80–0.84) [8–12]. Median PPV was 0.83 for studies including children only, 0.85 for studies including all ages, and 0.80 for studies including adults only.

The 19 studies that were excluded due to the small sample size (1–19 patients) presented PPV ranging from 0.40 to 1.00, with a median PPV of 0.90 (IQR 0.83–0.98) [31–49] (S2 Table). Including these studies in the meta-analysis, except for the one study with only one patient, did not change the model-based estimate of PPV (0.83, 95% confidence interval 0.79–0.86), but heterogeneity increased ($I^2$ = 68%, p = 0.011).

Regarding the effects of morphological properties of the abscess on PPV, 9 studies (39%) reported on rim enhancement (Table 2) [8, 14, 16, 18, 19, 21, 24, 26, 30], and 10 studies (43%) reported on abscess diameter [8, 10, 12, 14, 16, 19, 21, 22, 24, 26]. The meta-regression analysis found no overall effect of the moderators (CT criteria for an abscess, size of collection, rim enhancement, pediatric patients, or publication age) (p = 0.306). Regarding technical details of CT acquisition, 26% of the included studies reported on CT manufacturer and model, 22% on

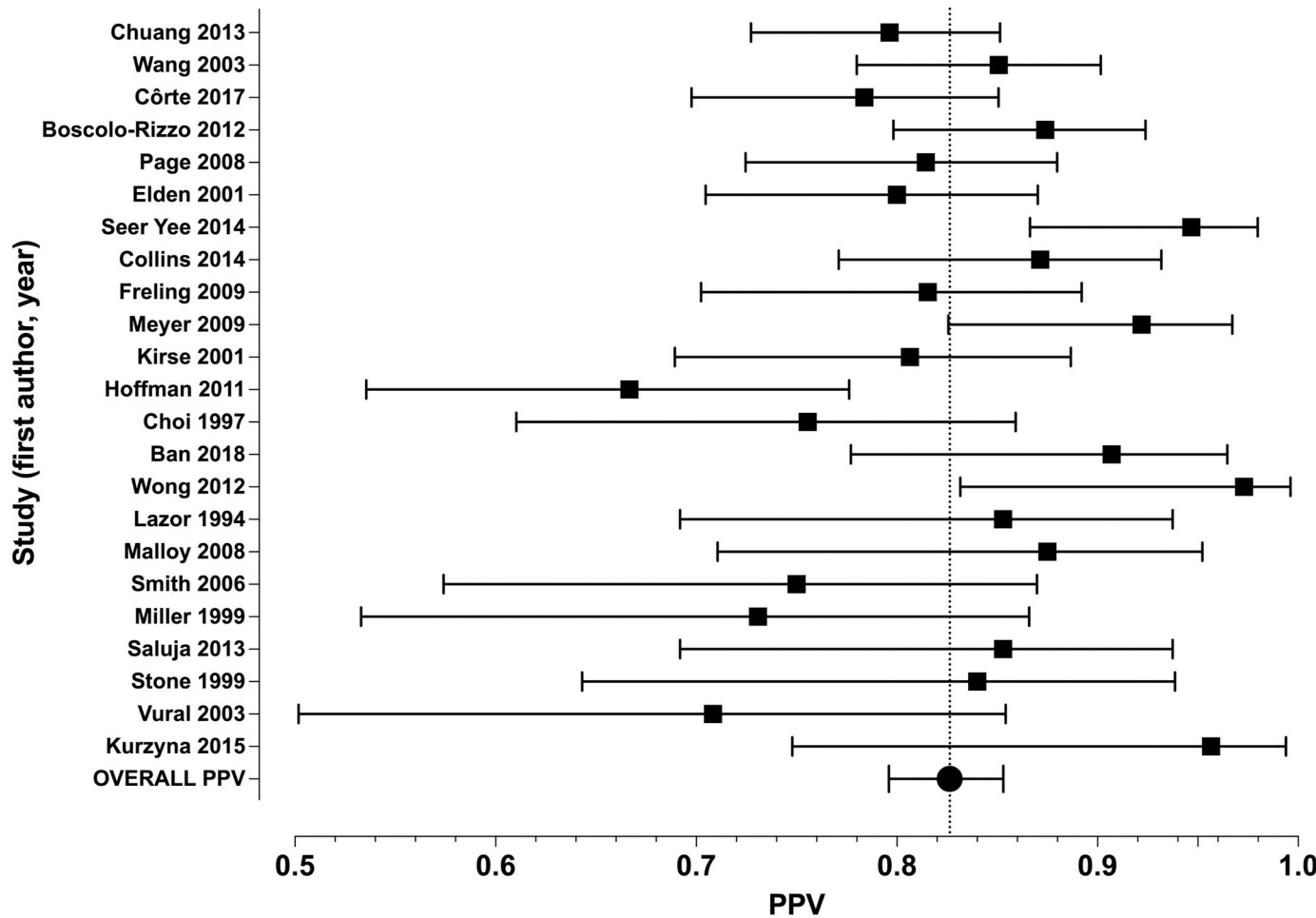

**Fig 2. Forest plot of PPV values from individual studies and the model-based pooled estimate of PPV.** Bars represent 95% confidence intervals.

slice thickness, 17% on contrast injection volume, 13% on contrast injection rate, and 9% on scanning delay after contrast injection (S3 Table).

## Discussion

We found an overall PPV of 0.83 for the diagnosis of a neck abscess using contrast-enhanced CT. Although publication bias was not evident in the funnel plot, studies with smaller sample sizes tended to show larger variation in PPV. For example, the five studies including more than 90 patients all have PPV in the range of 0.78–0.85 [8–12], whereas studies with less than 70 patients have a wider range of 0.67–0.97 [16–30]. We found evidence for considerable heterogeneity across the studies. Meta-regression did not reveal any significant modulation by reporting on morphological factors (such as size and ring enhancement). Overall, the current study highlights the difficulty of detecting drainable abscesses with contrast-enhanced CT and cautions against relying solely on individual morphological criteria.

A recent prospective study on retropharyngeal abscesses in children highlights the difficulties in correctly diagnosing drainable abscesses [27]. In that study, purulence was surgically demonstrated in 29 out of 34 patients (85%) triaged as having an "abscess" (clear central hypodensity, clear ring enhancement, scalloping of abscess wall). However, purulence was also

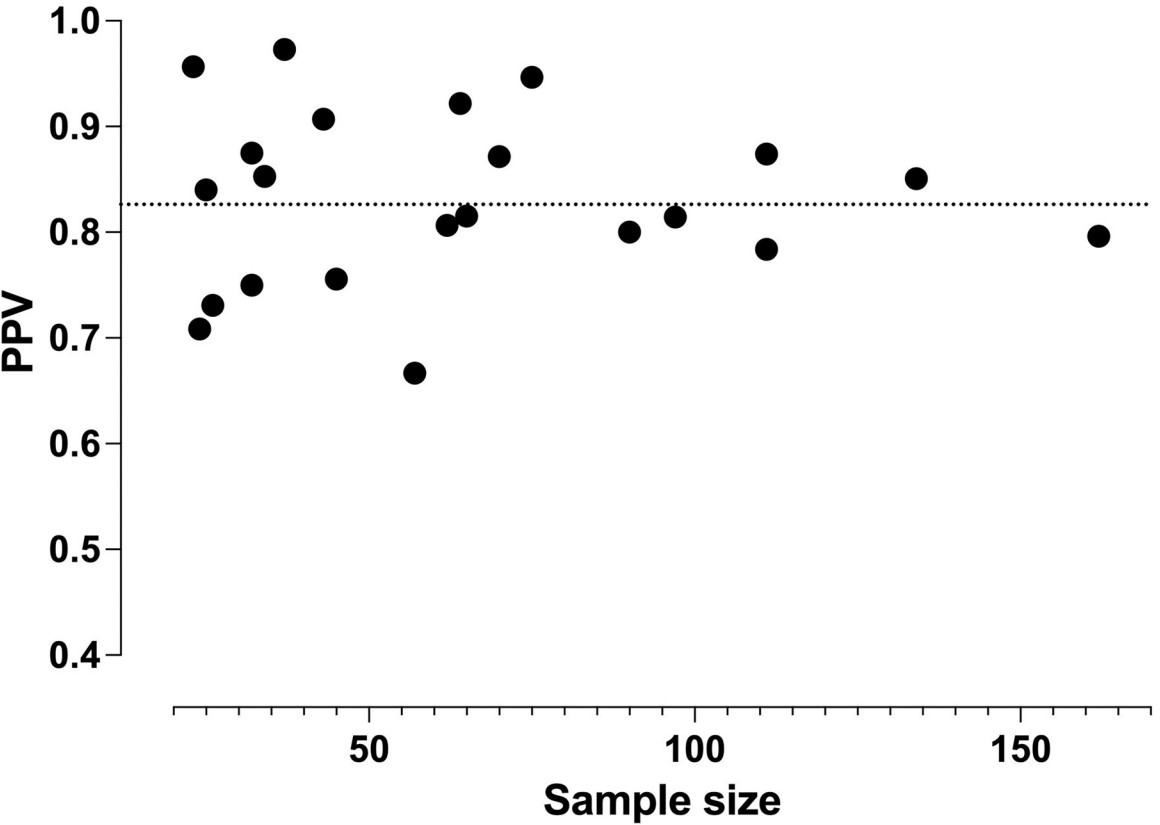

**Fig 3. Funnel plot of PPV values from individual studies plotted over study sample size.** The dotted line represents the model-based pooled estimate of PPV (0.83).

found in 29 out of 39 patients (74%) triaged as having a "phlegmon" (central hypodensity with or without ring enhancement, but not as prominent as with frank pus; round or oval process, not scalloped [27]. Thus, the differentiation between surgically drainable and non-drainable collection with these somewhat ambiguous and subjective CT criteria may be challenging.

**Table 2. Prevalence of rim enhancement (RE) of abscesses in studies that had reported on RE.**

| Author | Subpopulation | Ring enhancement (RE) | | | |
|---|---|---|---|---|---|
| | | Complete | Incomplete | Some[a] | None |
| Chuang | | 66% | | 66% | 34% |
| Seer Yee | All | 67% | 33% | 100% | |
| | Only TP | 70% | 30% | 100% | |
| Freling | All | 38% | 44% | 82% | 18% |
| | Only TP | 42% | 43% | 85% | 15% |
| Kirse | | 92% | | 92% | 8% |
| Hoffman | | 97% | | 97% | 3% |
| Ban | | 82% | | 82% | 18% |
| Malloy | | 5% | 86% | 91% | 9% |
| Miller | | 55% | | 55% | 45% |
| Kurzyna | | 43% | 57% | 100% | |

[a]At least some RE, either complete or complete plus incomplete, if reported separately. FP, false positive; TP, true positive.

This study has several limitations. Most importantly, almost all studies were retrospective in nature. In addition, most studies were not primarily designed at examining the diagnostic accuracy of CT imaging, but rather included radiological and surgical information as part of a comprehensive description of patient cohorts. Radiological and surgical information were not always well described. Many studies suffered from a small sample size. We specifically assessed the risk of bias and applicability concerns using QUADAS-2 criteria [5]. Regarding patient selection, patients seemed to have been randomly selected, although we found an applicability concern because many studies included only children. Regarding the index test, CT was always done before surgery, although not all studies reported surgery within 48 hours of imaging. Further, not all studies reported CT criteria for an abscess. Theoretically, newer studies may have benefited from improved CT technology, although we did not find a correlation between PPV and the year of publication. Details of the CT acquisition, such as volume and injection rate of the contrast agent and delay after injection, were reported so rarely that their impact could not be systematically assessed. Regarding the reference standard, surgical reports may have been incomplete, ambiguous, or even erroneous, and subject to interpretation. Surgeons may have had different levels of competence, and surgical detection of small abscesses may have been difficult. Finally, not all patients with a positive index test (abscess on CT) had reference standard (surgical proof). Taken together, many confounding factors limit the interpretation and practical utility of this meta-analysis. Yet, PPV data from larger studies seem to converge reasonably around the overall estimate.

Six studies reported sensitivity and specificity [12, 14, 15, 26, 28, 29], and two additional studies reported sensitivity alone [20, 23]. Sensitivity ranged from 0.68 to 0.99, and specificity from 0.18 to 0.63 (Table 1). Accurate estimation of sensitivity and specificity requires a full 2x2 diagnostic table. Due to partial verification bias, patients are much more likely to undergo surgery (reference standard) after a positive index test (abscess on CT) than after a negative index test (no abscess on CT). Therefore, the proportions of true and false negatives are not well known, and sensitivity and specificity will be biased. Because of this limitation, we performed the meta-analysis on PPV. Some studies have provided evidence for true and false negatives. For example, a fairly large study on pediatric neck abscesses found 8 false negative cases among 17 surgically treated CT-negative patients, suggesting a negative predictive value (NPV) of 0.53 [13]. Other NPV values reported in the literature are 0.89 [26], 0.70 [17], 0.53 [29], and 0.44 [28], whereas some studies reported zero false negatives [16, 20]. False negative findings on CT may delay appropriate surgical treatment, but an accurate assessment of the magnitude of this problem is difficult because of partial verification bias.

Abscesses may be more challenging to detect in some areas than in others, both surgically and on CT. A recent large study found a PPV of 0.91 for abscesses extending to multiple spaces, 0.87 for submandibular abscesses, 0.80 for parapharyngeal abscesses, and only 0.50 for retropharyngeal abscesses [8]. Regarding specific anatomical locations, most studies included patients with various locations. Five studies focused on pediatric retropharyngeal abscesses and reported a median PPV of 0.81 (range 0.67–0.85) [12, 18, 19, 27, 28].

Characteristic features of an abscess are a low-density core with surrounding rim enhancement, the presence of air, and irregularity of the collection wall [2]. However, there remains uncertainty in these diagnostic variables in the differentiation of abscess from phlegmon. Low-density core, which is measured as Hounsfield units (HU), is usually considered consistent with an abscess. However, HU values were not significantly different between phlegmon and abscess in one study [25], and another study reported a difference between drainable (HU 18) and non-drainable (HU 25) collections [21]. Another study reported that HU<32 was seen in 85% of abscesses and 40% of phlegmons [19], indicating a large overlap between the two conditions. One study found even higher HU in drainable than in non-drainable collections [14].

Thus, HU values gathered from included studies indicate too much overlap between values of abscesses and phlegmons to make definite recommendations for reliable cut-offs for clinical practice.

Rim enhancement is the formation of a granulomatous capsule around the infection, and it is considered one of the characteristic features, especially in the later stage of abscesses. Rim enhancement may be partial or complete. However, about two-thirds of false positive collections also had rim enhancement (Table 2), suggesting low specificity. Due to partial verification bias, the sensitivity of RE is arguably overestimated as most false negative cases remain undiscovered. Complete rim enhancement in late stages of abscess formation may explain some of the false negatives [28, 29]. Indeed, complete rim enhancement has been reported to be present in 42–82% (average 61%) of patients with surgically confirmed abscesses [14, 16, 21, 26, 30] (Table 2). Overall, on average 88% of patients with surgically confirmed abscesses have been reported to present either partial or complete rim enhancement on CT (Table 2). PPV has been reported higher in collections with rim enhancement (0.85) than in those without (0.67) [16]. One study found the rims of all radiologically defined abscesses only 5% complete enhancing, 86% partially enhancing, and 9% not enhancing [24], and another study found 45% no rim enhancement in abscesses [26]. Rim enhancement may also be present in a large proportion of non-purulent collections [8, 16, 18, 19, 21]. Seven studies reported rim enhancement in 14–100% of surgically confirmed non-purulent collections. On average, these studies reported 64% of non-purulent collections to have rim enhancement, although it is unclear whether these presented partial or complete enhancement [8, 14, 16, 18, 19, 26, 30].

All studies reporting on size found large collections more likely to be abscesses than small collections. Similarly, high PPV for multi-space abscesses is probably explained by the large diameter of the collection, while detection of abscesses in anatomically narrow spaces may be difficult. Thus, a recommendation for differentiating an abscess from a phlegmon based on size remains difficult due to variability in reported cutoff values. Some studies compared the sizes of true versus false positives whereas others compared medically treated versus operated collections. Also, the methods of determining the size of the collection varied (largest diameter, cross-sectional area, volume). In a large study, PPV was 0.73 for lesions of 0.9–3.0 cm in diameter, and 0.87 for lesions larger than 3.0 cm in diameter, with false positives being smaller (2.4 cm) than true positives (3.4 cm) [8]. A relatively large study found that all false positive abscesses were smaller than 3.5 cm in diameter [16]. Regarding the prediction of surgery, abscesses less than 2 cm in diameter were less likely to be operated on than those more than 2 cm in diameter in another study [10]. Yet another study found that surgically treated abscesses were larger (3.3 cm) than medically treated abscesses (2.5 cm) and that abscesses larger than 2 cm and 3 cm predicted the failure of medical treatment and repeat surgery, respectively [19]. Three studies analyzed the cross-sectional area or volume of the abscess: collections with an area larger than 2 $cm^2$ were more likely to be true positives (89%) than those with an area less than 2 $cm^2$ (65%) [12]; surgically (12.4 $cm^3$) and medically (7.2 $cm^3$) treated abscesses were not statistically significantly different [24]; and false positive abscesses were smaller (2.2 $cm^3$) than all collections (6.8 $cm^3$) [26].

Air formation was reported in four studies included in the final analysis [8, 14, 16, 21]. Altogether, only 6% (7/127) of negative radiological and surgical findings and 23% (66/292) of abscesses presented with air. Despite the infrequence of the finding, when present, the air was found to be a reliable predictor of pus. Two relatively large studies found PPV of 0.93 and 0.91 for lesions with air [8, 14]. In another study, air always indicated the presence of abscess [16]. Thus, the presence of air seems to be a highly specific, yet insensitive predictor of drainable abscesses.

Scalloping, or irregular morphology of the abscess wall, was found to be a useful predictor of purulence in a study that reported 94% PPV for scalloping findings on CT [18]. Also,

another study found a 96.9% PPV for "rim irregularity" [14]. This probably represents a late stage of abscess formation. One study, including 22 true positive abscesses and one false positive abscess, found scalloping in 39% of the abscesses, and the remaining abscesses were described as regular [30]. Thus, irregularity of the collection wall indicated purulence with high reliability.

Regarding the contrast agent, there is currently no consensus on the optimal amount or speed of administration or the delay between administration and imaging. A recent study found high PPV (0.92) for CT with a biphasic mode of contrast infection and a single scan phase [50]. The biphasic injection consists of a double mode of injection with single scanning. The first part of contrast is injected slowly, and it is expected to improve interstitial contrast concentration in the soft tissues (impregnation phase). This is supposed to better enhance the center of a phlegmon in contrast to the hypodense abscess core. The second rapid injection phase (vascular phase) should enhance vascular and other soft tissue structures. The biphasic contrast injection is suggested to allow better delineation of the abscess walls of different maturations and thus enhance differentiation. However, the superiority of this method lacks scientific studies in comparison to conventional single-phase contrast injections [50].

A potential limitation of CT in diagnosing drainable abscesses is imaging at an early stage of abscess formation, when an enhancing granulomatous capsule may not yet be present. This limitation is reflected in the variability of rim-enhancement in abscesses on CT imaging as referenced in greater detail above. Another challenging clinical scenario is the infected lymph node: differential diagnosis between non-suppurative lymphadenitis, suppurative lymphadenitis, and extra-nodal frank abscesses may be difficult. This limitation may be most pronounced in children, in whom lymphadenitis is common [19].

Despite the limitations of individual morphological criteria for an abscess, CT is a well-established modality for neck emergencies in clinical practice. In many instances, CT can detect the source and extent of the disease, assess potential complications, and provide critical information for surgical planning. Moreover, surgical decision-making is also based on potential complications including mediastinal involvement, airway compromise, laboratory findings, and overall clinical assessment–not solely on radiological detection and measurement of an abscess.

## Alternative methods: Spectral CT and magnetic resonance imaging (MRI)

Spectral CT is a novel application of CT, in which the attenuation data of multiple different energy spectra is being acquired, enabling a more accurate assessment of tissue attenuation even the separation of specific materials (such as water, iodine, and calcium) [51]. Spectral CT can be obtained by source-based (such as dual source, or rapid kilovolt switching) or detector-based techniques, the latter including the most advanced method, photon-counting CT. Dual-energy CT (DECT) is a subset of spectral CT, in which the spectra of two peak energies are obtained. DECT is the most common clinical application of spectral CT and has the potential to offer improved soft tissue sensitivity in head and neck disease compared with traditional single-energy CT [52, 53]. Specifically, DECT seems to enhance the detection of head and neck cancer [54], but evidence for the utility of DECT in neck infections is still limited. One study showed that a lower tube voltage of 80-kilovolt peak improved the delineation of peritonsillar abscesses compared to the typical 120-kilovolt peak tube voltage [55], suggesting that DECT might have better PPV for neck abscesses than single-energy CT. Another study found improved delineation of abscesses using 40-kiloelectron volt virtual monochromatic images (VMIs) and iodine maps compared with conventional 120-kilovolt peak images [56].

Magnetic resonance imaging (MRI) is not based on tissue attenuation of ionizing radiation and has superior soft tissue discrimination compared with CT [57]. The diagnostic criteria for

an abscess in MRI are straightforward: an abnormal T2-hyperintense, non-enhancing collection with a low apparent diffusion coefficient from diffusion-weighted imaging, surrounded by abnormal tissue enhancement [58]. Defined this way, the diagnosis of an abscess has substantial interobserver reliability. PPV of MRI for diagnosing surgically confirmed abscesses is very high (0.95) [58–62], which is clinically meaningful, because many patients with positive abscess finding on imaging undergo surgery. To put these numbers from both modalities in perspective, a PPV of 0.83 for CT may result in unnecessary surgery in approximately 1 in 6 patients, compared with 1 in 20 patients for MRI. Thus, more accurate imaging of neck infections will likely benefit the patients. MRI has the added benefit of accurately describing soft tissue edema patterns, that have prognostic value [59, 61, 62].

## Conclusions

The positive predictive value (PPV) of contrast-enhanced CT for neck abscesses is 0.83. Morphological criteria, such as low-density core, rim enhancement, and scalloping, are often applied, but studies show considerable overlap between drainable and non-drainable collections. Despite these limitations, CT is a well-established modality for neck emergencies in clinical practice. Alternative methods, such as spectral CT or MRI, may provide improved soft tissue discrimination.

## Supporting information

**S1 Checklist. PRISMA 2009 checklist.**
(PDF)

**S1 Table. QUADAS-2 assessments.**
(DOCX)

**S2 Table. Studies excluded due to small sample size.**
(DOCX)

**S3 Table. Technical details of CT acquisition in the included studies.**
(DOCX)

## Author Contributions

**Conceptualization:** Kimmo Mattila, Jussi Hirvonen.

**Data curation:** Jon Hagelberg, Bernd Pape, Jaakko Heikkinen, Janne Nurminen, Jussi Hirvonen.

**Formal analysis:** Jon Hagelberg, Bernd Pape, Jaakko Heikkinen, Janne Nurminen, Jussi Hirvonen.

**Funding acquisition:** Jussi Hirvonen.

**Investigation:** Jon Hagelberg, Bernd Pape, Jaakko Heikkinen, Janne Nurminen, Jussi Hirvonen.

**Methodology:** Kimmo Mattila, Jussi Hirvonen.

**Project administration:** Jussi Hirvonen.

**Resources:** Jussi Hirvonen.

**Supervision:** Jussi Hirvonen.

**Writing – original draft:** Jon Hagelberg.

**Writing – review & editing:** Bernd Pape, Jaakko Heikkinen, Janne Nurminen, Kimmo Mattila, Jussi Hirvonen.

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
