## [Decision Letter · Decision Letter 0]

27 Sep 2022

PONE-D-22-24583Diagnostic accuracy of CT in neck infections: a systematic review and meta-analysis of positive predictive valuePLOS ONE

Dear Dr. Hirvonen,

Thank you for submitting your manuscript to PLOS ONE. After careful consideration, we feel that it has merit but does not fully meet PLOS ONE’s publication criteria as it currently stands. Therefore, we invite you to submit a revised version of the manuscript that addresses the points raised during the review process.

We look forward to receiving your revised manuscript.

Kind regards,

Essam Al-Moraissi

Academic Editor

PLOS ONE

3. Please upload a copy of Supporting Information Table S1, S2 and S3 which you refer to in your text on page 7 and 9.

Reviewers' comments:

Reviewer's Responses to Questions

**Comments to the Author**

1. Is the manuscript technically sound, and do the data support the conclusions?

Reviewer #1: Yes

Reviewer #2: Yes

2. Has the statistical analysis been performed appropriately and rigorously? 

Reviewer #1: Yes

Reviewer #2: Yes

3. Have the authors made all data underlying the findings in their manuscript fully available?

Reviewer #1: Yes

Reviewer #2: Yes

4. Is the manuscript presented in an intelligible fashion and written in standard English?

Reviewer #1: Yes

Reviewer #2: Yes

5. Review Comments to the Author

Reviewer #1: This is a meta-analysis and systematic review of contrast enhancement CT in neck infection.

I have several comments on this study.

1. At first, I recommend authors to add "contrast enhancement" in the title.

2. More details are needed in table 1 - age - mean? or median? or range? from each study

3. In meta-regression, publication year can be added, as there might be a difference according to CT.

4. Authors need to analyze more details, for example, subgroup analysis according to age groups (children? adults?)

5. There is not a statistical analysis in size of collection, but description of previous studies. If there is no statistical analysis, these can be moved to discussion section. and this applies to other subsections of results as well.

6. In some sentences, authors mention that "there were 5 studies included in this analysis" but they did not cite them properly.

Reviewer #2: This is a well-written meta-analysis on the diagnostic accuracy of CT for neck infections. I have only minor remarks:

a) The title does not accurately reflects the purpose of this study. Merely using the terms 'neck infection' is too vague. As the authors themselves state, 'To be included, studies had to specifically report having included “abscesses” and not simply “infections”, because the latter might include non-purulent conditions such as phlegmon/cellulitis. Moreover, in several points it is clear that the main objective here is accuracy for abscess. Please, adopt a more specific, clearer title.

b) What is the rationale behind excluding studies with less than 20 patients? Please address this in your text. I am curious regarding the results of this meta-analysis if these studies were not excluded, considering that they encompass 206 patients excluded (a significant number in a review including 1453 patients). Did the authors try to see the results if these patients were included? Would these results differ too much from the current ones?

6. PLOS authors have the option to publish the peer review history of their article (what does this mean?). If published, this will include your full peer review and any attached files.

Reviewer #1: No

Reviewer #2: No

---

## [Author Response · Author response to Decision Letter 0]

30 Sep 2022

Author response

We thank the reviewers for their comments and questions concerning our manuscript. We greatly appreciate this opportunity to improve our paper. As described below, we have now responded to each point, and think that the manuscript is significantly improved. Please, find our responses below highlighted in blue font. We have organized the responses according to reviewer number and question/comment number (e.g., R1-1, R1-2, …, R2-1, R2-2, ...). New or revised text in the manuscript is indicated by the “Track changes” function. In addition, some typos were corrected.

Reviewer #1: This is a meta-analysis and systematic review of contrast enhancement CT in neck infection.

I have several comments on this study.

R1-1: At first, I recommend authors to add "contrast enhancement" in the title.

Authors’ reply R1-1: We agree, and have now included the term “contrast-enhanced” in the title. We have also made the title more specific by mentioning “abscesses” rather than “infections” (please also see point R2-1). The revised title now reads as follows:

“Diagnostic accuracy of contrast-enhanced CT for neck abscesses: a systematic review and meta-analysis of positive predictive value”

R1-2: More details are needed in table 1 - age - mean? or median? or range? from each study

Authors’ reply R1-2: We agree, and have now added the mean (or median) ages as well as their ranges, as reported by the original publications.

R1-3: In meta-regression, publication year can be added, as there might be a difference according to CT.

Authors’ reply R1-3: We agree and have now added publication age in years (2022 minus the year of publication) as a term in the meta-regression analysis, along with the factor of whether the study was done in children only (see point R1-4). However, the effect of the moderators was not significant. This is now explained in the Methods section, page 7, as follows:

“Meta-regression analyses were carried out with the following moderators: whether CT criteria of abscess were defined (yes/no), whether the size of the collection was considered (yes/no), whether rim enhancement was being analyzed (yes/no), whether the study was done only in children (yes/no), and age of publication (years, 2022 minus year of publication)”

New results are given in the Results section, page 10, as follows:

“The meta-regression analysis found no overall effect of the moderators (CT criteria for an abscess, size of collection, rim enhancement, pediatric patients, or publication age) (p=0.306).”

In summary, including the year of the publication into the meta-regression model did not significantly change the results.

R1-4: Authors need to analyze more details, for example, subgroup analysis according to age groups (children? adults?)

Authors’ reply R1-4: We agree, and have now analyzed different age groups separately. We found that median PPV was very similar in the groups, as now explained in the Results section, page 9:

“Median PPV was 0.83 for studies including children only, 0.85 for studies including all ages, and 0.80 for studies including adults only.”

In addition, we analyzed the effect of focusing only on children as a term in the meta-regression analysis but including this term did not have a significant effect on the model-based PPV. For the revised text, see point R1-3 above.

In summary, the overall PPV appears quite similar among studies including different age groups.

R1-5: There is not a statistical analysis in size of collection, but description of previous studies. If there is no statistical analysis, these can be moved to discussion section. and this applies to other subsections of results as well.

Authors’ reply R1-5: We agree, and have moved all these descriptive subsections from the Results section to the Discussion section. For clarity, some subtitles were removed, along with redundant text that was present in both the Results and Discussion sections in the previous version of the manuscript.

R1-6: In some sentences, authors mention that "there were 5 studies included in this analysis" but they did not cite them properly.

Authors’ reply R1-6: We have now provided appropriate citations to these claims.

Reviewer #2: This is a well-written meta-analysis on the diagnostic accuracy of CT for neck infections. I have only minor remarks:

R2-1: a) The title does not accurately reflects the purpose of this study. Merely using the terms 'neck infection' is too vague. As the authors themselves state, 'To be included, studies had to specifically report having included “abscesses” and not simply “infections”, because the latter might include non-purulent conditions such as phlegmon/cellulitis. Moreover, in several points it is clear that the main objective here is accuracy for abscess. Please, adopt a more specific, clearer title.

Authors’ reply R2-1: We agree, and have now included the term “abscess” rather than “infections” in the title. We have also added “contrast-enhanced” as suggested by Reviewer #1 (please also see point R1-1). The revised title now reads as follows:

“Diagnostic accuracy of contrast-enhanced CT for neck abscesses: a systematic review and meta-analysis of positive predictive value”

R2-2 b) What is the rationale behind excluding studies with less than 20 patients? Please address this in your text. I am curious regarding the results of this meta-analysis if these studies were not excluded, considering that they encompass 206 patients excluded (a significant number in a review including 1453 patients). Did the authors try to see the results if these patients were included? Would these results differ too much from the current ones?

Authors’ reply R2-2: The rationale was to exclude small studies, in which an absolute difference in just a few patients might have a disproportionally large impact on PPV. Indeed, we noted that the median PPV among these smaller studies was somewhat higher (0.90) than that among the larger studies included (0.84). 

We also performed the meta-analysis including these smaller studies (except for the one study with only one patient). We found that the model-based PPV estimate did not change (0.83, 95% confidence interval 0.79–0.86), but heterogeneity increased (I2=68%, p=0.011). This suggests that these smaller studies do not significantly contribute to the overall estimate of PPV, but add heterogeneity. These results are presented in the Results section, page 9.

This is consistent with the fact that even among the included studies with >20 patients, studies with smaller samples tend to have higher variability in PPV. This is presented in Fig. 3. and referred to in the Results section, page 9.

---

## [Decision Letter · Decision Letter 1]

10 Oct 2022

Diagnostic accuracy of contrast-enhanced CT for neck abscesses: a systematic review and meta-analysis of positive predictive value

PONE-D-22-24583R1

Dear Dr. Hirvonen,

We’re pleased to inform you that your manuscript has been judged scientifically suitable for publication and will be formally accepted for publication once it meets all outstanding technical requirements.

Kind regards,

Essam Al-Moraissi

Academic Editor

PLOS ONE

Additional Editor Comments (optional):

Dear Authors, both reviewers accept your manuscript in the current status.

Reviewers' comments:

Reviewer's Responses to Questions

**Comments to the Author**

1. If the authors have adequately addressed your comments raised in a previous round of review and you feel that this manuscript is now acceptable for publication, you may indicate that here to bypass the “Comments to the Author” section, enter your conflict of interest statement in the “Confidential to Editor” section, and submit your "Accept" recommendation.

Reviewer #1: All comments have been addressed

Reviewer #2: All comments have been addressed

2. Is the manuscript technically sound, and do the data support the conclusions?

Reviewer #1: Yes

Reviewer #2: Yes

3. Has the statistical analysis been performed appropriately and rigorously? 

Reviewer #1: Yes

Reviewer #2: Yes

4. Have the authors made all data underlying the findings in their manuscript fully available?

Reviewer #1: Yes

Reviewer #2: Yes

5. Is the manuscript presented in an intelligible fashion and written in standard English?

Reviewer #1: Yes

Reviewer #2: Yes

6. Review Comments to the Author

Reviewer #1: Authors addressed all the comments I made previously. This is a well-written manuscript, and, I recommend "accept" for this study.

Reviewer #2: My comments were addressed. Thank you.

In the first paragraph of Discussion ('For example, the five studies more than 90 patients all have PPV in the range of 0.78–0.85'), I think that there is a word missing after 'studies'. This can be corrected during proof review.

7. PLOS authors have the option to publish the peer review history of their article (what does this mean?). If published, this will include your full peer review and any attached files.

Reviewer #1: No

Reviewer #2: No

---

## [Editor Report · Acceptance letter]

17 Oct 2022

PONE-D-22-24583R1 

Diagnostic accuracy of contrast-enhanced CT for neck abscesses: a systematic review and meta-analysis of positive predictive value 

Dear Dr. Hirvonen:

I'm pleased to inform you that your manuscript has been deemed suitable for publication in PLOS ONE. Congratulations! Your manuscript is now with our production department. 

Kind regards, 

on behalf of

Dr. Essam Al-Moraissi 

Academic Editor

PLOS ONE